# Conformal Prediction for Molecular Properties under Label Shift

**Hyeonsu Lee**[1][*]   **Juyeon Kim**[1][*]   **Erkhembayar Jadamba**[1][*]   **Seungjin Choi**[2]   **Hyunjin Shin**[1][†]

[1]MOGAM Institute for Biomedical Research    [2]Intellicode

{hyeonsu.lee, juyeon.kim.m, erkhembayar.jadamba, hyunjin.shin}@mogam.re.kr
seungjin@intellicode.co.kr

## Abstract

Drug discovery and development underpins healthcare but remains costly and failure-prone. A critical bottleneck lies in predicting molecular properties such as solubility, potency, and toxicity, which directly determine whether a candidate can advance from preclinical to clinical trials. Artificial Intelligence (AI) has accelerated this process, yet its reliability is often undermined by distribution shift, as experimental conditions frequently diverge from training data. In addition, conventional point predictions provide only single-value estimates, offering limited guidance for high-stakes experimental design. We address these challenges with a conformal prediction framework tailored to label shift. By weighting conformal scores using marginal label probability ratios, our method produces statistically rigorous prediction intervals without retraining. This enables robust uncertainty quantification even when property distributions drift, directly tackling one of the most pervasive obstacles to applying AI in real-world drug development. By moving beyond accuracy alone to provide actionable confidence measures, our approach enhances the trustworthiness of AI-driven predictions. This further aligns predictive modeling with regulatory demands for transparency and uncertainty reporting and ultimately supports more reliable decision-making in billion-dollar development pipelines.

## 1   Introduction

Drug discovery and development is characterized by prolonged timelines, substantial resource requirements, and a high likelihood of failure. More specifically, developing and bringing a single new drug to market can cost from $314 million to $2.8 billion and take over a decade, but failure rates can reach up to larger than 90% across the entire development life cycle [1]. This remarkable inefficiency underscores the growing importance of artificial intelligence (AI) because it can substantially reduce the need for chemical and biological experiments by making predictions on molecular properties such as solubility, bioavailability, or toxicity. However, the current performance of AI for this purpose does not appear to be fully compelling to drug development specialists. A key component contributing to this insufficient performance is the uncertainty embedded in prediction by AI, mainly originating from data characterization and model training. In response to this limitation, recent FDA guidance (2025) for artificial intelligence in drug and device development explicitly requires that AI systems should provide "**appropriate confidence intervals**" and "**uncertainty estimates**" when supporting regulatory submissions [2, 3]. This requirement indicates that actively considering the uncertainty, thereby improving the reliable predictions, will be a critical part of AI applications to drug discovery and development.

---

[*]Equal contribution
[†]Corresponding author

The reliability of AI predictions is often undermined by the problem of distribution shift [4], a scenario where test data differ substantially from the training data. This is particularly an urgent issue in drug discovery, where novel compounds frequently occupy chemical spaces unseen during training. The consequence is often summarized as overconfident yet unreliable predictions, which is an unacceptable risk when billions of dollars and patient outcomes are at stake.

To address this gap, reliable estimation of uncertainty is essential. Conformal prediction, also known as conformal inference, is a versatile and statistically principled framework that constructs prediction intervals around model outputs [5, 6, 7]. Its foremost advantage is its distribution-free and finite-sample validity, which guarantees that prediction intervals will contain the true label with a user-specified probability (e.g., 90%), regardless of dataset size or underlying distributional assumptions. This property represents a major improvement over many traditional statistical methods [7, 8]. Originally pioneered by Vladimir Vovk and his colleagues in the 1990s, the core mechanism involves a simple calibration step where a small holdout dataset is used to convert an arbitrary heuristic notion of uncertainty from a pre-trained model into a rigorous one, typically by computing conformal scores and their empirical quantiles [5, 7, 9].

This methodology is broadly applicable across various machine learning tasks, ranging from image classification to regression. It has also been significantly extended to address complex real-world challenges such as covariate shift, distribution drift, and the control of general risks. As a result, it has become an indispensable tool for reliable uncertainty quantification in high-stakes applications [6, 7]. In drug discovery, this unreliability often stems from two specific types of distribution shift: covariate shift [6] and label shift [10]. Covariate shift occurs when the distribution of molecular structures $P(x)$ changes between training and test. For instance, in drug discovery, covariate shift is observed when a model trained on diverse chemical libraries is applied to a new and more specialized set of molecules.

On the other hand, label shift, the primary target of this work, arises when the distribution of the target property $P(y)$ changes, while the conditional distribution of features given the label $P(x \mid y)$ remains invariant [11, 12, 13]. This situation commonly arises when research priorities shift toward discovering molecules with property values underrepresented in the original training data, such as compounds exhibiting exceptionally high potency or low toxicity. Although various machine learning solutions have been developed to address distribution shifts [6, 10, 14, 15, 16] label shift remains relatively underexplored, particularly in continuous regression-based tasks that are frequently encountered in molecular property prediction[11, 13, 17].

Addressing issues related to distribution shifts is becoming increasingly important as more sophisticated AI models, such as large language models (LLMs) pretrained on large-scale chemical databases [18, 19, 20], are applied to explore the complex relationships between molecular structure and function. However, designing novel molecules with these AI models inherently requires highly accurate predictions under distribution shift. In general, AI models trained under the assumption of identically distributed data often fail to account for label shift, as this assumption typically leads to overconfident yet incorrect single-value predictions for novel and unseen molecules. This underscores the necessity of uncertainty quantification, and highlights that the estimated uncertainty must be integrated with AI predictions to produce realistic and robust prediction intervals, even for state-of-the-art LLM-based AI models.

In response to these challenges, we propose a new framework that generates reliable prediction intervals for molecular properties, even under significant label shift. Our method builds on conformal prediction, a machine learning technique that provides distribution-free and finite-sample guarantees on prediction intervals [6, 21]. Standard conformal prediction assumes exchangeability between training and test data, an assumption violated under label shift. To address this issue, we develop a scheme based on *weighted conformal prediction*. In our framework, the corrective weights are derived from the ratio of the target to the source label distributions, which we estimate using versatiel approaches such as black box shift estimation (BBSE) [11], regularized learning under label shifts (RLLS) [12], and maximum likelihood estimation (MLE) [13].These techniques enhance the practicality of our approach, as the label shift can be directly estimated from the outputs of both unbiased and biased predictive models.

In conclusion, our method effectively mitigates the adverse effects of label shift without requiring costly model retraining. It generates statistically rigorous prediction intervals that adapt to changing property distributions, thereby providing a more realistic assessment of a molecule's potential.

Overall, this work makes a key contribution to the development of reliable AI for drug discovery by offering a robust methodology that ensures models remain trustworthy when navigating the uncertain frontiers of novel chemical space.s

## 2 Methods

The overall design of our framework is illustrated in Figure 1. The pipeline consists of the following steps: (i) the base prediction model is trained using source training set to perform predictions in the label shift environment, (ii) to quantify label shift, importance weights, which represent the ratio of the target domain's marginal label distribution to the source domain's marginal label distribution, are estimated using weight set through methods such as BBSE, RLLS, and MLE, (iii) nonconformity scores, such as absolute residuals, are computed for each data point in calibration set based on the predictions of the trained model and their actual labels, and (iv) the weighted quantile of the nonconformity scores is calculated by incorporating the estimated importance weights, which is then used to construct statistically valid prediction intervals under label shift for new test points. This high-level schema highlights how our approach adapts standard conformal prediction to remain valid under label shift.

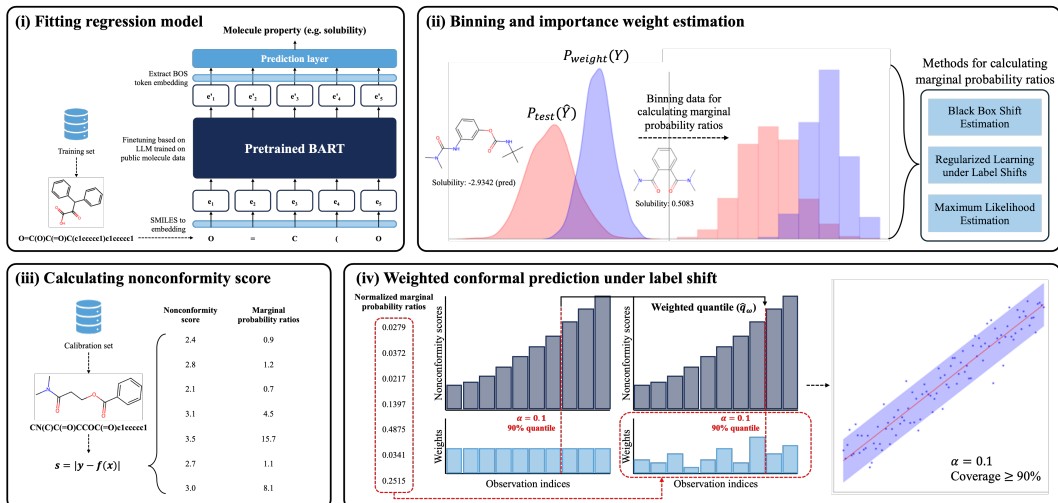

Figure 1. Schematic diagram of conformal prediction for molecular properties under label shift.

We now formalize our approach for conformal prediction under label shift.

### 2.1 Problem Formulation

Let the source data be $\mathcal{D}_s = \{(x_i, y_i)\}_{i=1}^n$, where $x_i \in \mathcal{X}$ is a molecular representation and $y_i \in \mathbb{R}$ is the continuous property of interest. These data points are drawn from a distribution $p(x, y)$. We are also given a set of unlabeled data from a target domain, $\mathcal{D}_t = \{x_j\}_{j=n+1}^{n+m}$, drawn from a different distribution $q(x, y)$.

The label shift assumption posits that the conditional distribution of features given the label remains constant across domains, while the marginal label distribution changes:

$$p(x|y) = q(x|y) \quad \text{and} \quad p(y) \neq q(y) \tag{1}$$

Our goal is to construct a prediction interval, $C(x_{\text{test}})$, for a new test input $x_{\text{test}}$ from the target domain that satisfies the marginal coverage guarantee at a desired confidence level $1 - \alpha$:

$$\mathbb{P}(y_{\text{test}} \in C(x_{\text{test}})) \geq 1 - \alpha \tag{2}$$

### 2.2 Binning and Importance Weight Estimation

To apply classification-based shift estimation techniques, we first discretize the continuous response variable $y$ into $K$ bins, creating a pseudo-label $\tilde{y} = \text{bin}(y) \in \{0, 1, \cdots, K-1\}$ These pseudo-labels

are used only for estimating weights. The discretization was performed using equally sized bins, and
the bin range was determined by the minimum and the maximum values of the data used to calculate
the marginal probability ratios.

The importance weight for each bin is defined as the ratio of the target and source pseudo-label
probabilities:

$$w(\widetilde{y}) = \frac{q(\widetilde{y})}{p(\widetilde{y})} \tag{3}$$

In practice, the source probabilities $p(\widetilde{y})$ are calculated from the empirical frequencies in a held-out
portion of the source data. The target probabilities $q(\widetilde{y})$ are estimated from the unlabeled target data
using methods like BBSE, RLLS, or MLE, which leverage the outputs of a model trained on the
binned source data.

Since MLE could not directly estimate $q(\widetilde{y})$ from predictions, we adopted a probabilistic approach.
For each sample, we modeled a Gaussian centered at the prediction with standard deviation equal to
the root mean squared error (RMSE) from the weight set. The probability of the sample falling into a
bin was then given by the cumulative distribution function (CDF) difference at the bin's bounds. Any
negative probabilities arising from numerical errors were set to zero, and the resulting probability
vector was normalized to ensure that its elements summed to one.

## 2.3 Weighted Conformal Prediction under Label Shift

To ensure the statistical validity of our method, we partition the source data $\mathcal{D}_s$ into three disjoint
subsets, preventing data leakage between steps:

1. Proper training set ($\mathcal{D}_{\text{train}}$): Used to train the base prediction model, $f$.
2. Weights set ($\mathcal{D}_{\text{weight}}$): Used to estimate the label shift importance weights.
3. Calibration set ($\mathcal{D}_{\text{cal}}$): Used to compute nonconformity scores and calibrate the prediction
   intervals.

The weighted conformal prediction algorithm then proceeds as follows:

1. For each point $(x_i, y_i)$ in the calibration set $\mathcal{D}_{\text{cal}}$, compute a nonconformity score. For
   regression, this is typically the absolute residual:

$$s_i = |y_i - f(x_i)| \tag{4}$$

2. Assign the corresponding estimated importance weight $\widehat{w}_i = \widehat{w}(\widetilde{y}_i)$ to each score $s_i$, where
   $\widetilde{y}_i$ is the bin of the true label $y_i$.
3. Compute the *weighted quantile* $\widehat{q}_w$ from the set of scores $\{s_i\}$ and weights $\{\widehat{w}_i\}$. This
   quantile is the value that satisfies:

$$\widehat{q}_w = \inf \left\{ s : \frac{\sum_{i=1}^{n_{cal}} \widehat{w}_i \cdot \mathbb{I}\{s_i \leq s\}}{\sum_{j=1}^{n_{cal}} \widehat{w}_j} \geq 1 - \alpha \right\} \tag{5}$$

4. For a new test point $x_{\text{t}}$, the final *prediction interval* is formed by centering the weighted
   quantile around the model's point prediction:

$$C(x_{\text{t}}) = [f(x_{\text{t}}) - \widehat{q}_w, \quad f(x_{\text{t}}) + \widehat{q}_w] \tag{6}$$

By using this weighted quantile, the method corrects for the distributional shift and restores the
marginal coverage guarantee under the label shift assumption.

# 3 Experimental Settings

## 3.1 TDC Solubility AqsolDB

Solubility AqSolDB [22] from the Therapeutics Data Commons (TDC) [23], which provides mea-
surements of compound solubility in aqueous solutions. This dataset serves as a benchmark for
studying molecular physicochemical properties and for developing predictive models of drug solubil-
ity. AqSolDB specifically provides solubility information, which is a critical factor in drug design and
delivery systems, and consists of 9,982 compounds. For each compound, experimentally measured
logarithmic solubility values ($logS$) and molecular structure information are included.

## 3.2 Chemical Large Language Model Finetuning

The large language model (LLM) employed in this study is based on a BART [24] architecture and has been optimized for the analysis of chemical data. The model was pretrained utilizing approximately 200 million unlabeled SMILES (Simplified Molecular Input Line Entry System) [25] data collected from Chembl [26], PubChem [27], ZINC [28], Enamine [29], Coconut [30], and Drugbank [31]. Through this pretraining, the model acquired enriched representations specific to the chemical domain, encompassing approximately 250 million parameters. The fine-tuning process was conducted via full fine-tuning of the pretrained LLM.

## 3.3 Data Splitting

We conducted a conformal prediction simulation utilizing split conformal prediction methods to address continuous label shift. The objective of this simulation was to ensure that the marginal probability ratio-based weights are "exchangeable" between the nonconformity score distributions of the training and test datasets when the label distribution $Y$ differs between them. By guaranteeing this exchangeability, the constructed prediction intervals satisfy a minimum coverage of $1 - \alpha$ in a distribution-free manner.

The experiment was repeated 1,000 times, and in each iteration, the original data is divided into two subsets: source data $\mathcal{D}_s$ and target data $\mathcal{D}_t$. The two subsets are split in a 60% to 40% ratio of the total data. Here, $\mathcal{D}_s$ is divided into three subsets ($\mathcal{D}_{train}$, $\mathcal{D}_{weight}$, $\mathcal{D}_{cal}$) of equal size. $\mathcal{D}_t$ is split into two subsets ($\mathcal{D}_{no\_shift}$, $\mathcal{D}_{shift}$) to evaluate coverage performance under label shift conditions and without label shift. $\mathcal{D}_{no\_shift}$ represented 50% of $\mathcal{D}_t$ and corresponded to test data without label shift. $\mathcal{D}_{shift}$ is generated by sampling with replacement from $\mathcal{D}_t$, excluding $\mathcal{D}_{no\_shift}$. During this sampling process, the probability of selecting each data point was proportional to a specific weight, where $w(y) = \exp(y^T \beta)$. These weights were assigned based on the magnitude of $y$.

## 4 Experimental Results

**Split conformal prediction fails under label shift**   As expected, the traditional split conformal prediction method exhibited a significant decline in coverage performance on the label-shifted test set compared to the non-shifted test set. In Figure 2, the coverage distribution for the shifted data (red) is notably shifted to the left relative to the distribution for the non-shifted data (gray), with the average coverage falling substantially below the nominal target. These findings highlight the limitations and unreliability of standard uncertainty quantification methods under label shift conditions.

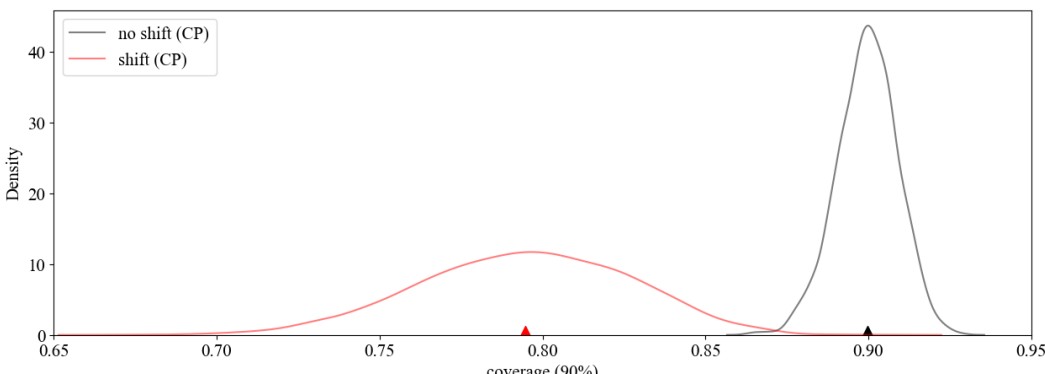

Figure 2. The KDE distributions for 1000 repeated experiments using standard split conformal prediction and the proposed methods are shown. The triangular markers on the x-axis correspond to the mean value of each distribution. The gray and red lines indicate the coverage distributions obtained by applying standard split conformal prediction to the test data without and with label shift (non-uniformly subsampled), respectively.

**Ratios of marginal probabilities recover coverage loss from label shift**   Our proposed method, which integrates weighted conformal prediction (WCP) with marginal probability ratios estimated via

BBSE, RLLS, and MLE with bias-corrected temperature scaling (BCTS), provides statistically valid and reliable prediction intervals for addressing continuous label shift problems. The experimental results demonstrate that the proposed approach achieves improved predictive coverage compared to the traditional split conformal prediction (CP) method. Specifically, WCP consistently outperformed CP in terms of average coverage, as evidenced by the coverage distribution shown in Figure 3. The coverage distribution of WCP shifted to the right relative to CP, indicating that WCP more frequently generated prediction intervals that included the true labels. This allowed WCP to effectively achieve the desired coverage level, even under label shift conditions. These results provide empirical validation that weighted conformal prediction, when its weights are derived from marginal probability ratios, can produce more robust and reliable prediction intervals. This approach proves especially effective in addressing challenges presented by label shift scenarios.

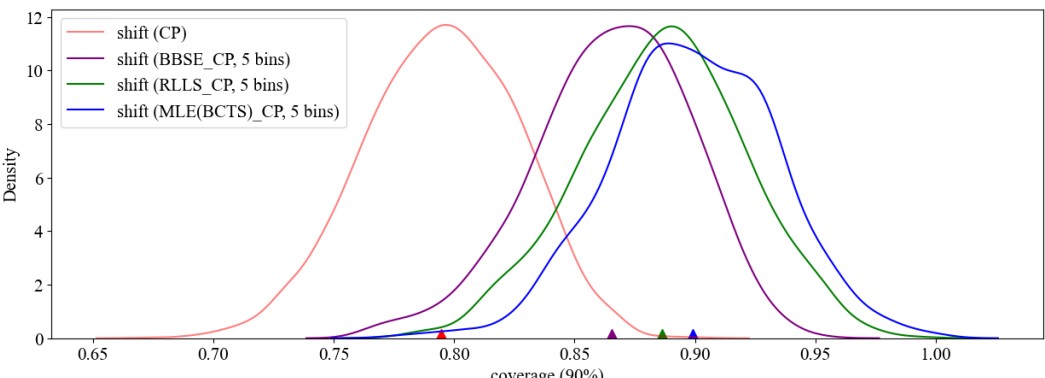

Figure 3. The KDE distributions for 1000 repeated experiments using standard split conformal prediction and the proposed methods are shown. The triangular markers on the x-axis correspond to the mean value of each distribution. Coverage distributions are shown for the test data under label shift: the red line corresponds to standard split conformal prediction, while the purple, green, and light blue lines correspond to conformal prediction with weights calculated via BBSE, RLLS, and MLE (BCTS), respectively.

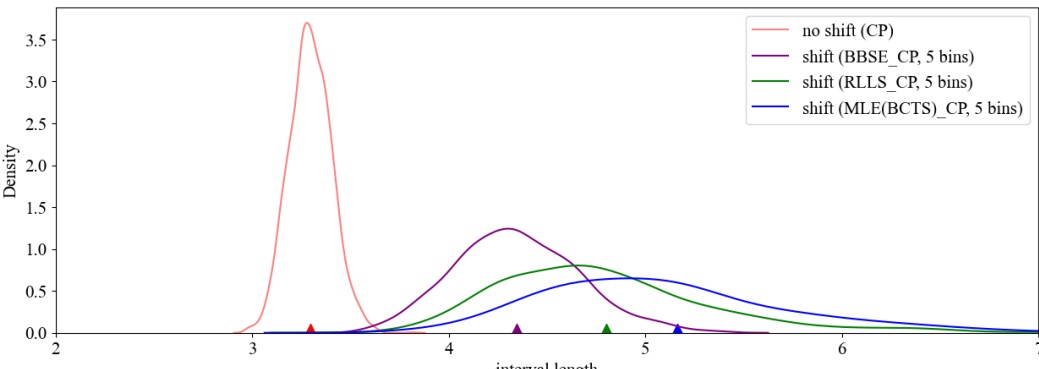

Figure 4. The KDE distributions for 1000 repeated experiments using standard split conformal prediction and the proposed methods are shown. The triangular markers on the x-axis correspond to the mean value of each distribution. Interval length distributions are shown for the test data under label shift: the red line corresponds to standard split conformal prediction, while the purple, green, and light blue lines correspond to conformal prediction with weights calculated via BBSE, RLLS, and MLE (BCTS), respectively.

**Approaches for estimating marginal probability ratios** When comparing the performance of BBSE, RLLS, and MLE, we observed that MLE achieved the highest coverage, followed by RLLS and then BBSE (Figure 3). Moreover, MLE showed robust performance in coverage recovery with respect to the number of bins (Table 2). This improvement can be explained by two factors: first, the use of

bias-corrected calibration reduces systematic bias across classes; and second, the MLE algorithm benefits from a theoretical guarantee of convergence to a global optimum [13]. Nevertheless, this improvement in coverage came with certain trade-offs. The prediction intervals generated by MLE were generally wider (Figure 4), whereas BBSE and RLLS produced relatively narrower intervals.

# 5   Limitation

Our approach requires splitting source data into training, weighting, and calibration sets, which can reduce effective training size and hurt performance in low-sample regimes. Data augmentation methods [32, 33] may mitigate this. Moreover, standard conformal intervals are suboptimal for heteroscedastic data; techniques such as Conformalized Quantile Regression (CQR) [21] could provide more adaptive intervals. Exploring these directions remains future work.

# 6   Summary

This paper presents a practical and statistically grounded framework for producing reliable prediction intervals for molecular property prediction under label shift. By weighting conformal prediction with estimates of the target label distribution—obtained via BBSE, RLLS, and MLE—our method restores the coverage guarantees that split conformal prediction loses under distribution shift. When tested on the AqSolDB dataset with a large-scale pretrained chemical language model, our weighted conformal prediction consistently achieves more robust coverage than traditional approaches, with no need for costly retraining. The method is compatible with various estimation techniques, while maximum likelihood–based corrections achieve the best performance in coverage recovery. Our key contribution lies in developing a generalizable and model-agnostic framework that addresses an essential gap in the reliability of molecular property prediction. By ensuring statistically rigorous uncertainty quantification under label shift, our approach advances AI-based drug discovery toward regulatory compliance and real-world adoption, ultimately increasing confidence in high-stakes decisions on which compounds progress through the development pipeline.

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

## A  Detailed Experimental Settings

All models were trained on NVIDIA A100 SXM4 40GB GPUs. As the model requires approximately 7.5 GB of memory for training, it is also feasible to run it on GPUs with lower specifications. Detailed model hyperparameters are represented in Table 1.

Table 1. Hyperparameters for training BBSE, RLLS, and MLE

| Hyperparameter | BBSE | RLLS | MLE |
|---|---|---|---|
| Optimizer | AdamW [34] | AdamW [34] | AdamW [34] |
| Adam betas | (0.9, 0.99) | (0.9, 0.99) | (0.9, 0.99) |
| Learning rate | 5e-5 | 5e-5 | 5e-5 |
| Weight decay | 0.1 | 0.1 | 0.1 |
| Warmup steps | 100 | 100 | 100 |
| Error rate $\alpha$ | 0.1 | 0.1 | 0.1 |
| Batch size | 16 | 16 | 16 |
| Max length | 150 | 150 | 150 |
| Label shift $\beta$ | -0.5 | -0.5 | -0.5 |
| BART hidden dim | 768 | 768 | 768 |
| Predictor hidden dims | [512, 256] | [512, 256] | [512, 256] |
| Calibration | None | None | BCTS |
| Epochs | 10 | 10 | 10 |

## B  Additional Results

**Coverage recovery performance based on the number of bins**   We varied the number of bins in BBSE, RLLS, and MLE, and for each configuration, the mean and standard deviation of coverage and interval length were reported over 1000 trials (Tables 2). The characteristics of the coverage distribution vary with the number of bins used in applying WCP through label discretization. Specifically, with fewer bins, certain trials exhibited high coverage, but the overall coverage distribution showed greater variance. Conversely, as the number of bins increased, the variance of the overall coverage distribution decreased, resembling the distribution observed when CP was applied to a non-shifted test dataset. This phenomenon can be interpreted as follows: with fewer bins, the coarse discretization of nonconformity scores leads to unstable quantile estimation, causing irregular fluctuations in the length of prediction intervals and significantly increasing the variance of the overall coverage distribution. On the other hand, as the number of bins increases, the estimation errors caused by discretization are reduced, resulting in a more stable and narrower (relatively) coverage distribution.

Table 2. Comparison of average coverage and interval length across different bin numbers for BBSE, RLLS, and MLE methods.

| Bins | BBSE | | RLLS | | MLE (BCTS) | |
|---|---|---|---|---|---|---|
| | Coverage | Length | Coverage | Length | Coverage | Length |
| 5 | $0.865_{\pm0.033}$ | $4.346_{\pm0.316}$ | $0.886_{\pm0.034}$ | $4.804_{\pm0.590}$ | $0.899_{\pm0.034}$ | $5.163_{\pm0.832}$ |
| 15 | $0.845_{\pm0.032}$ | $3.993_{\pm0.236}$ | $0.877_{\pm0.032}$ | $4.577_{\pm0.42}$ | $0.895_{\pm0.034}$ | $5.031_{\pm0.620}$ |
| 50 | $0.804_{\pm0.033}$ | $3.414_{\pm0.149}$ | $0.859_{\pm0.032}$ | $4.204_{\pm0.297}$ | $0.892_{\pm0.034}$ | $4.947_{\pm0.581}$ |
| 100 | $0.782_{\pm0.033}$ | $3.165_{\pm0.123}$ | $0.843_{\pm0.032}$ | $3.934_{\pm0.222}$ | $0.858_{\pm0.031}$ | $4.192_{\pm0.242}$ |

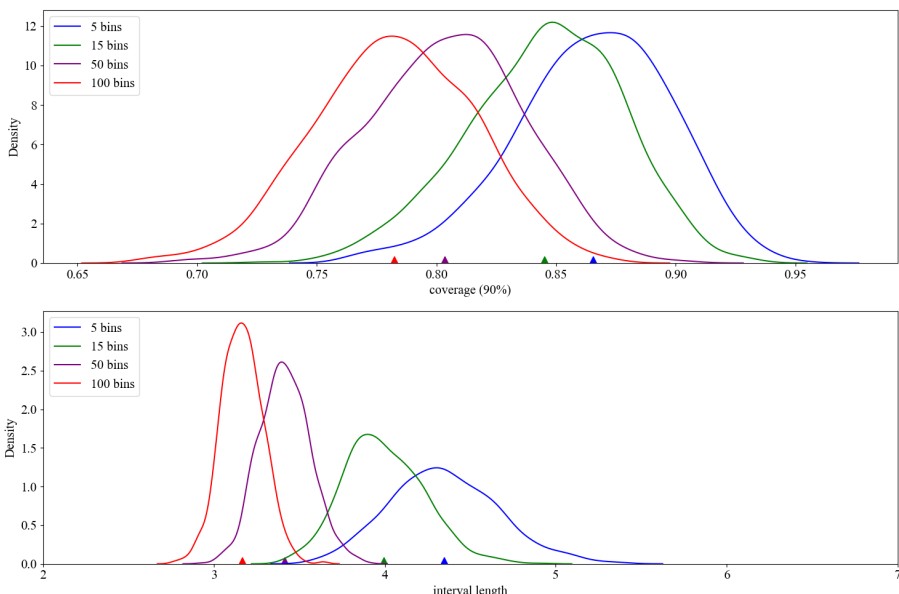

Figure 5. The KDE coverage distribution (top) and interval length distribution (bottom) for 1000 repeated experiments using standard split conformal prediction and the proposed methods are shown. Each color represents the number of bins used to calculate marginal probability ratios via BBSE. The triangular markers on the x-axis indicate the mean value of each distribution.

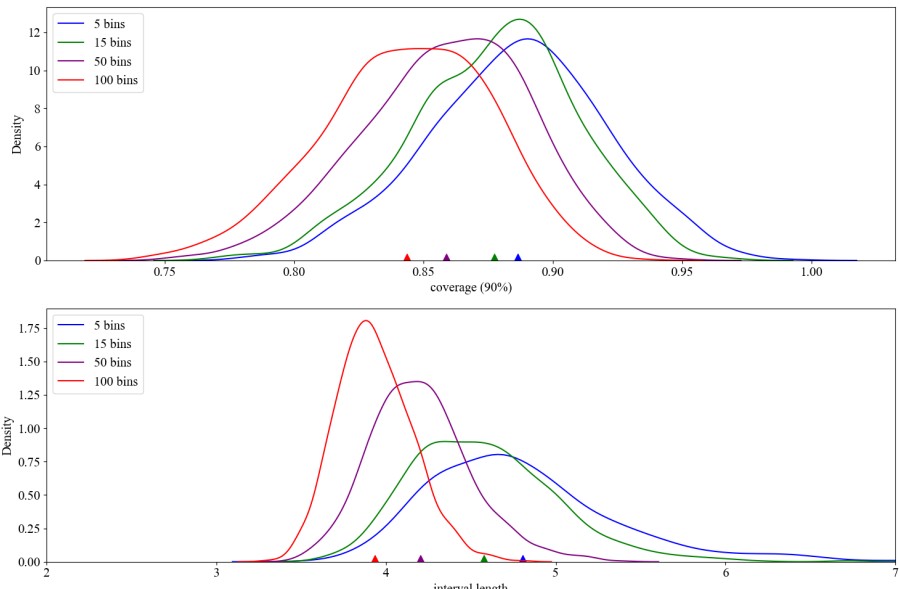

Figure 6. The KDE coverage distribution (top) and interval length distribution (bottom) for 1000 repeated experiments using standard split conformal prediction and the proposed methods are shown. Each color represents the number of bins used to calculate marginal probability ratios via RLLS. The triangular markers on the x-axis indicate the mean value of each distribution.

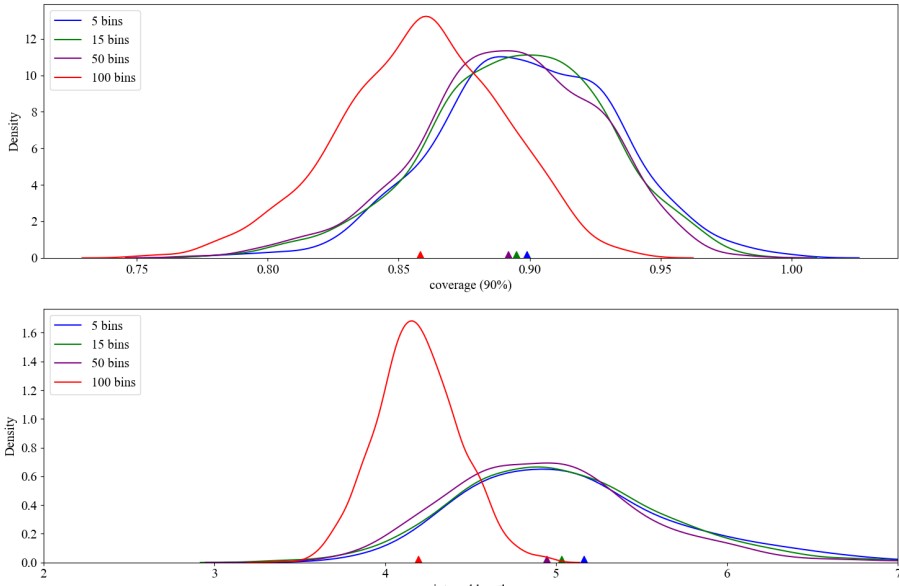

Figure 7. The KDE coverage distribution (top) and interval length distribution (bottom) for 1000 repeated experiments using standard split conformal prediction and the proposed methods are shown. Each color represents the number of bins used to calculate marginal probability ratios via MLE. The triangular markers on the x-axis indicate the mean value of each distribution.

