# OpenReview forum: "Conformal Prediction for Molecular Properties under Label Shift"
_NeurIPS.cc/2025/Workshop/Reliable_ML — NeurIPS 2025 - Reliable ML Workshop_

### Official Review · Reviewer_QKUE · 2025-09-17
**Strong introduction and methodology for conformal prediction with label shift, but evaluation could be expanded.**

**Rating:** 7
**Confidence:** 3

**Review:**

Summary:
This paper studies conformal prediction under label shift, with an application to predicting molecular properties using large language models (LLMs). The focus is on label shift, where the distribution of the output labels differs between training and test data. The authors propose a framework that incorporates importance weights, estimated from the training and test distributions, into the conformal prediction process. This adjustment enables the method to maintain valid uncertainty guarantees even in the presence of label shift.

Strengths:
The paper provides a clear explanation of why label shift is an important problem in machine learning and its practical implications, particularly for molecular property prediction. The introduction to conformal prediction is thorough and accessible, making the method approachable for readers who may be unfamiliar with it. The use of diagrams, especially Figure 1, effectively illustrates the key ideas and helps readers understand the proposed approach.

Weaknesses / Limitations:
The main weakness lies in the presentation and interpretation of the results. While the authors argue that maximum likelihood estimation (MLE) performs best, the evidence provided seems somewhat limited. The differences between MLE and alternative methods like RLLS are relatively small, which raises questions about whether the conclusion is sufficiently supported. More detail or additional evaluation metrics would strengthen this aspect of the work.

Suggestions for Authors:
Figure 3 shows an irregular distribution for the MLE histogram, suggesting that more samples may be needed to obtain a better characterization. In addition, the claim that MLE performs best seems mainly based on the mean being closest to 90% (Figure 3) and the largest interval length (Figure 4). Since the results are close to RLLS, it would be useful to consider other metrics, such as the shape/spread of the histogram, to provide stronger evidence. Acknowledging this explicitly would help clarify the contribution. Finally, the paper could benefit from an explicit statement of what the inputs are in the molecular property prediction task. While it is clear that the outputs are molecular properties such as solubility or toxicity, the nature of the input data is less clearly specified and should be highlighted.

Ethics: N/A

---

### Official Review · Reviewer_iDkF · 2025-09-19
**Good paper on weighted conformal prediction enables reliable solubility prediction under label shift**

**Rating:** 7
**Confidence:** 3

**Review:**

Summary

The paper addresses the problem of unreliable uncertainty estimates in molecular property prediction, focusing on the challenge of label shift, where the distribution of target properties differs between training and test data. To tackle this, the authors propose a Weighted Conformal Prediction framework, extending standard conformal prediction to provide uncertainty intervals that remain valid under label shift. The method proceeds in four stages: (1) a base prediction model is trained on source data, (2) importance weights are estimated to correct for discrepancies between source and target label distributions, (3) nonconformity scores are computed on a calibration set, and (4) prediction intervals are constructed by taking a weighted quantile of these scores, thereby restoring valid coverage guarantees under distributional shift. The approach is evaluated on the AqSolDB solubility dataset. The authors demonstrate that standard split conformal prediction suffers a substantial loss of coverage, and their weighted method consistently restores the desired guarantees. The main result is that WCP yields statistically valid and reliable prediction intervals under label shift without requiring retraining of the base model, making it both practical and robust for high-stakes applications such as drug discovery.

Strengths:

The paper is relevant in tackling uncertainty quantification which is an important gap in drug discovery. The framework is model agnostic and adapts conformal prediction to BBSE, RLLS, MLE. Pipeline is clear and gives a step by step method to reproduce it

Weakness:

In real-world applications, distribution shifts would likely be combined with a feature/covariate shift? In the real world these distribution shifts are unlikely to be uncoupled. The current paper assumes these shifts are uncoupled, but this may not hold in practice and this is limiting the robustness of the proposed method.
The paper does not analyze whether the broad intervals generated by MLE remain practically useful or become too uninformative. A spot-check assessment between coverage reliability and interval efficiency would strengthen the work.

Suggestions for Authors:

The author should consider a comparison against other uncertainty quantification methods.
The authors should consider extending the evaluation to another ADMET property dataset to establish generality beyond solubility.
Authors should discuss how the method behaves if the label-shift assumption is violated, or if covariate shift is also present.